# OphthaDT: Generative Digital Twins for Forecasting Visual Acuity Trajectories in Ophthalmology

**Pietro Belligoli** [* 1 2]  **Nikita Makarov** [* 1 3 4]  **Sayedali Shetab Boushehri** [1]  **Fabian Schmich** [† 1]
**Raul Rodriguez-Esteban** [† 5]  **Michael Menden** [† 3 6]

## Abstract

Precision medicine in ophthalmology requires accurate longitudinal predictions, but the fragmented nature of multimodal clinical data remains a barrier to forecasting. We introduce OphthaDT, an LLM-based digital twin for ophthalmology that serializes longitudinal patient histories from 3,220 patients across four Phase III clinical trials into structured narratives to forecast best corrected visual acuity (BCVA). In benchmarks spanning up to 100 weeks, OphthaDT demonstrated the lowest prediction error in neovascular age-related macular degeneration (nAMD), achieving an average mean absolute error (MAE) reduction of 6.0% compared to all baselines. In diabetic macular edema (DME), OphthaDT demonstrated competitive performance against all baselines while outperforming Random Forest and XGBoost by an average MAE reduction of 2.6% and 6.9%, respectively. Results reveal that OphthaDT's predictive advantage scales with trajectory complexity: whereas linear models remain effective for the more stable treatment responses of DME, OphthaDT's capacity is better suited for capturing the high longitudinal variability of nAMD. Finally, OphthaDT handles irregular sampling without imputation, positioning LLM-based clinical trajectory modeling as a methodology that could reduce patient burden and accelerate drug development.

[*]Equal contribution [†]Equal supervision. [1]Computational Sciences Center of Excellence, Roche, Penzberg, Germany [2]Technical University of Munich, Munich, Germany [3]Computational Health Center, Helmholtz Munich, Munich, Germany [4]Department of Biology, Ludwig Maximilian University of Munich, Munich, Germany [5]Computational Sciences Center of Excellence, Roche, Basel, Switzerland [6]Department of Biochemistry and Pharmacology, Bio21 Molecular Science and Biotechnology Institute, The University of Melbourne, Parkville, Australia. Correspondence to: Raul Rodriguez-Esteban <raul.rodriguez-esteban@roche.com>, Michael P. Menden <michael.menden@unimelb.edu.au>.

*Proceedings of the $2^{nd}$ ICML Workshop on Foundation Models for Structured Data*, Seoul, South Korea. 2026. Copyright 2026 by the author(s).

## 1. Introduction

Neovascular age-related macular degeneration (nAMD) and diabetic macular edema (DME) are leading causes of severe vision loss worldwide (Flaxman et al., 2017), with treatment outcomes varying substantially across individuals (Amoaku et al., 2015; Rofagha et al., 2013). Accurate forecasting of these outcomes could enable personalized treatments and improved clinical trial design through patient simulation. Digital Twins (DTs) address this challenge by serving as virtual patient replicas that learn from longitudinal multimodal data to generate individualized health forecasts (Bordukova et al., 2024; Kamel Boulos & Zhang, 2021).

Foundation models have facilitated the development of DTs by providing a principled approach to holistically model heterogeneous clinical data (Hegselmann et al., 2026; Wornow et al., 2023; Steinberg et al., 2024), enabling longitudinal trajectory forecasting on EHR data (Guo et al., 2023; Hur et al., 2023). An emerging method involves serializing longitudinal patient histories into structured textual representations, enabling LLMs to forecast clinical trajectories while naturally accommodating missingness and irregular sampling (Makarov et al., 2026). While this approach has shown strong performance in oncology, it remains unclear whether such modeling generalizes to therapeutic areas with different disease dynamics. Ophthalmology provides a compelling test case to evaluate this approach. Its primary outcome, best corrected visual acuity (BCVA), exhibits longitudinal trajectories, and treatment in this domain relies on localized intravitreal injections rather than systemic therapies.

We introduce OphthaDT, an LLM-based digital twin model for ophthalmology trained on 3,220 patients from four Phase III clinical trials. The model transforms fragmented trial records (visual acuity, imaging biomarkers, treatment history) and clinical data into structured narratives to forecast BCVA trajectories. Our contributions are threefold: (1) a serialization method that maps multimodal ophthalmological events into a tokenizable narrative for LLM-based forecasting; (2) high accuracy in BCVA forecasting across extended horizons (up to 100 weeks), setting a new benchmark for nAMD; and (3) evidence that LLM-based trajectory modeling generalizes beyond oncology to ophthalmic endpoints.

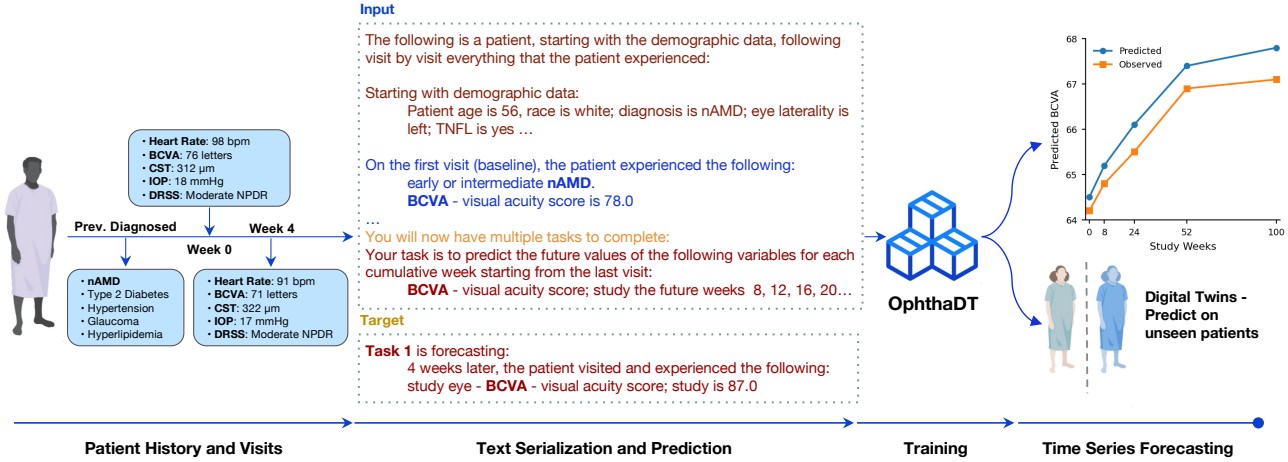

*Figure 1.* **OphthaDT pipeline overview.** (1) Patient history and baseline measurements are extracted from four Phase III clinical trials. (2) Longitudinal records are serialized into structured text prompts with forecasting instructions. (3) MedGemma 4B is fine-tuned on instruction–completion pairs. (4) The model predicts BCVA trajectories for unseen patients.

## 2. Related Work

**Digital Twins for Clinical Trajectory Forecasting.** Digital Twins offer a pathway to accelerate drug development by augmenting trials with virtual patient replicas. While most foundation models provide single-point outcomes, effective clinical decision-making requires comprehensive longitudinal trajectories (Pellegrini et al., 2026). Twin-Weaver (Makarov et al., 2026) introduced an open-source framework that builds DTs by serializing patient histories as text, jointly forecasting continuous biomarkers and discrete clinical events in oncology. However, the applicability of this generative paradigm to non-oncology settings with distinct functional endpoints remains underexplored.

**Trajectory Prediction in Ophthalmology.** Foundation models such as RETFound (Zhou et al., 2023), VisionFM (Qiu et al., 2024), and EyeCLIP (Shi et al., 2025) have advanced ophthalmic AI toward generalizable systems capable of diagnosis and disease prediction from retinal images. Whereas these models excel at image-based classification, they are typically limited to single-timepoint assessments and do not address longitudinal trajectory forecasting from structured clinical time series (Chia et al., 2024). At the patient trajectory level, prior work has used Random Forests and linear models to predict BCVA response (Rohm et al., 2018). However, these approaches often require manual feature engineering and lack the flexibility to accommodate the full breadth of multimodal longitudinal information.

## 3. Methods

We introduce OphthaDT, a model that forecasts BCVA trajectories from longitudinal patient histories in ophthalmology. OphthaDT addresses the challenge of heterogeneous and irregularly sampled clinical data through structured text representation (Fig. 1). By specializing this methodology for functional vision endpoints, OphthaDT enables long-term forecasting of BCVA.

**Notation** Let $\mathcal{P}$ be the set of patients, where each $p = (s_p, H_p) \in \mathcal{P}$ has static attributes $s_p$ (demographics and pre-trial medical history) and an observed history $H_p = [h_p^1, \ldots, h_p^n]$, where $h_p^i = $ (timestamp, event, value) with $n$ observations per patient. The prediction target $Y_p(t)$ denotes the set of future values to be forecasted from time $t$. In our setting, events include ophthalmological examinations, vitals, treatments, and adverse events. The next-token probability of the LLM is defined as $q_\theta(x_i \mid x_{1:i-1})$ for a given token sequence $x$, parameterized by $\theta$.

**Data** We utilized data from 3,220 patients with longitudinal records for nAMD and DME across four Phase III clinical trials. Each trial collected longitudinal records of disease progression and treatment interventions over scheduled clinical visits. Data were harmonized into trajectories integrating static baseline characteristics (demographics and medical history) with time-stamped clinical events, including vitals, treatments, examinations, and adverse events. The aggregated cohort was split at the patient level into 80% training, 10% validation, and 10% test sets, stratified by clinical trial and indication (nAMD/DME).

### 3.1. Text Serialization

OphthaDT encodes each patient's history by serializing it visit-by-visit into a structured narrative, mapping complex clinical events into a tokenizable format. The framework transforms the patient history $X_p(t)$ and prediction targets $Y_p(t)$ into a prompt organized into three hierarchical mod-

*Table 1.* **BCVA forecasting performance for nAMD and DME.** MAE is reported in ETDRS letters (lower is better); $R^2$ reflects the proportion of variance explained. OphthaDT is compared against Linear, RF, and XGBoost baselines. Best results per row are in **bold**.

| Indication | Timepoint | Linear Model | | Random Forest | | XGBoost | | OphthaDT | |
|---|---|---|---|---|---|---|---|---|---|
| | | MAE | $R^2$ | MAE | $R^2$ | MAE | $R^2$ | MAE | $R^2$ |
| nAMD | Week 8 | 6.19 | **0.59** | 6.53 | 0.55 | 6.82 | 0.51 | **6.13** | 0.58 |
| | Week 24 | 7.16 | 0.52 | 7.32 | 0.48 | 7.89 | 0.40 | **7.02** | **0.52** |
| | Week 52 | 8.55 | 0.39 | 8.81 | 0.39 | 9.26 | 0.34 | **8.34** | **0.40** |
| | Week 100 | 10.12 | **0.36** | 10.58 | 0.29 | 10.89 | 0.22 | **9.82** | 0.34 |
| DME | Week 8 | **5.61** | **0.54** | 6.03 | 0.47 | 6.23 | 0.43 | 5.62 | **0.54** |
| | Week 24 | **5.84** | **0.47** | 6.10 | 0.42 | 6.51 | 0.33 | 6.23 | 0.42 |
| | Week 52 | **7.17** | **0.32** | 7.54 | 0.24 | 7.76 | 0.22 | 7.37 | 0.29 |
| | Week 100 | **9.00** | **0.15** | 9.58 | 0.11 | 10.09 | 0.04 | 9.23 | 0.14 |

ules: (1) static baseline characteristics, comprising demographics and medical history (e.g., age, diagnosis, comorbidities); (2) longitudinal visit history, a chronological sequence containing BCVA scores, administered drugs, and adverse events; and (3) forecasting instructions, a numbered list of prediction tasks specifying the variables and future timepoints to be predicted. The output trajectories are encoded in a structured text format containing only the relevant output variables for the forecasted timepoints. The procedure for prompt construction is detailed in Algorithm 1, with a complete serialization example in Appendix A.4.

## 3.2. Forecasting Task

OphthaDT performs multi-horizon time-series forecasting of BCVA, the standard trial endpoint in ophthalmology measured on a 0–100 letter scale. The forecasting target $Y_p^{\text{forecast}}(t_{\text{forecast}})$ comprises future BCVA values up to a maximum horizon $t_{\text{forecast}} = t + \Delta t_{\text{forecast}}$, predicted at standardized intervals (cumulative study weeks 4, 8, 12, ..., 100).

**Trajectory Splitting.** For patient $p$, we define the input as the combination of static baseline data and longitudinal history up to a split time $t$: $X_p(t) = (s_p, H_p(t))$, with $H_p(t) = [h_p^i \mid h_p^i \in H_p, \ h_p^{i,\text{timestamp}} \leq t]$. In this work, we primarily focus on forecasting from treatment initiation ($t = 0$) to evaluate long-term predictive capacity, but the framework supports splitting at any arbitrary timepoint.

## 3.3. Training

The OphthaDT training pipeline uses supervised fine-tuning on longitudinal clinical data. We selected MedGemma 4B (Sellergren et al., 2026) as the backbone model for its biomedical pretraining, instruction-tuning capabilities, and open-source availability. We fine-tuned this model on the serialized instruction-completion pairs described above.

Each narrative is prepended with a system prompt (Appendix A.5) defining the model's forecasting role. The LLM

is trained using the standard cross-entropy loss, masked such that the gradient is computed only on the target completion tokens:

$$\mathcal{L}(\theta) = -\sum_{i=k}^{|x|} \log q_\theta(x_i \mid x_{1:i-1}) \qquad (1)$$

where $k$ is the index of the first target token and $x$ is the concatenation of the prompt and the target. Key hyperparameters used in fine-tuning are shown in Appendix A.2.

## 3.4. Inference

During inference, we approximate the model's predictive distribution by sampling $M = 10$ independent completions $\{\hat{Y}_p^m\}_{m=1}^M$ from $q_\theta$. Each text-based completion is decoded into a numerical trajectory by a structured parser that extracts the BCVA values from the generated tokens. To produce a stable estimate and mitigate sampling variance, the final forecast $\hat{Y}_p^{\text{forecast}}$ is calculated as the element-wise mean of these decoded trajectories across all forecasted timepoints:

$$\hat{Y}_p^{\text{forecast}} = \frac{1}{M} \sum_{m=1}^M \hat{Y}_p^m \qquad (2)$$

## 3.5. Baselines and Evaluation

We benchmarked OphthaDT against a linear model, Random Forest, and XGBoost, representing established baselines for outcome prediction in ophthalmology (Kikuchi et al., 2024). To ensure a fair comparison, all baselines utilized the same feature set as OphthaDT, with missing values addressed via mean imputation and categorical variables through one-hot encoding.

The primary evaluation metric was the mean absolute error (MAE), measured in early treatment diabetic retinopathy study (ETDRS) letters. In clinical practice, the ETDRS letter score is the gold standard for quantifying BCVA, where

a change of 5 to 15 letters is considered clinically meaningful (Beck et al., 2007; Csaky et al., 2008). In addition, we used the coefficient of determination ($R^2$) to quantify the proportion of variance captured by each model. Performance was assessed at four clinically significant milestones, as defined by our clinical experts: weeks 8, 24, 52, and 100. Baseline hyperparameters are detailed in Appendix A.3.

## 4. Results

OphthaDT was benchmarked against linear, Random Forest, and XGBoost models at four clinical milestones. Detailed performance metrics are provided in Table 1.

**nAMD**  In the nAMD cohort, OphthaDT achieved the lowest MAE across all forecasted timepoints, outperforming every evaluated baseline. Across the full 100-week horizon, the model demonstrated a mean error reduction of 5.7% relative to Random Forest and 2.1% relative to the linear model, averaging 6.0% across all three baselines. Although the linear model achieved comparable or slightly higher $R^2$ at several timepoints, OphthaDT provided more accurate point estimates (MAE) of BCVA throughout the forecast horizon.

**DME**  Results in the DME cohort indicate the effectiveness of linear models for trajectories with predictable treatment responses. This is consistent with evidence that treated eyes in DME converge toward a stable visual acuity plateau (Dugel et al., 2016). In this indication, the linear model attained the lowest MAE and the highest $R^2$ across the majority of timepoints, particularly during the initial induction phase at week 8. While the linear model remained the strongest baseline, OphthaDT demonstrated competitive performance by achieving a mean reduction in MAE of 6.9% over XGBoost and 2.6% over Random Forest across all timepoints.

**Subgroup Analysis**  Stratified analysis confirms that OphthaDT's performance is comparable across demographics and treatment cohorts (Fig. 2). In both indications, the model maintained similar predictive precision across treatment options, with DME additionally showing stable performance across patients with and without intact baseline retinal layers (TNFL: Yes/No). Although gender subgroups show some variance at intermediate timepoints, the overall error stability across different treatments suggests serialization captures the underlying clinical trajectories regardless of study protocol.

## 5. Conclusion

We introduced OphthaDT, an LLM-based digital twin model for ophthalmology that forecasts BCVA trajectories in

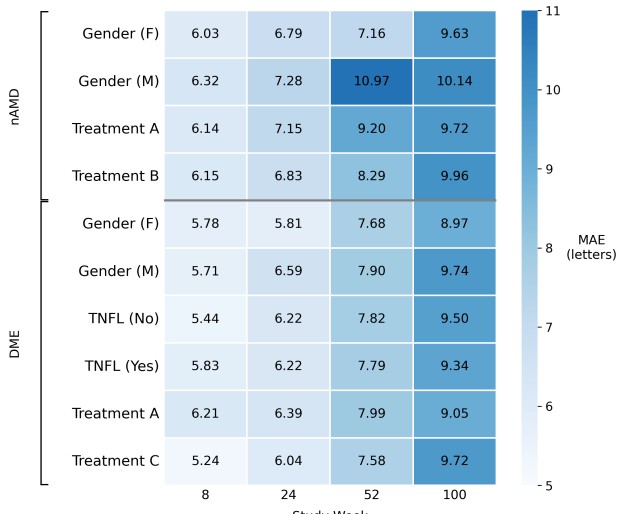

*Figure 2.* **Stratified forecasting performance across clinical subgroups.** Mean absolute error (in letters) for nAMD and DME. Rows correspond to demographic (age, gender) and clinical (treatment arm, TNFL) strata; columns to weeks 8, 24, 52, 100.

nAMD and DME by serializing fragmented clinical records into structured narratives. This approach achieved the lowest prediction error across all timepoints in nAMD and remained competitive with established baselines in DME.

The divergence in model ranking between nAMD and DME reveals a relationship between trajectory complexity and predictive advantage. In nAMD, where visual acuity is characterized by high volatility and acute fluctuations, OphthaDT's non-linear modeling capacity enabled it to achieve the lowest MAE across all timepoints. This aligns with evidence that nAMD outcomes involve non-linear biomarker interactions that traditional models struggle to capture (Schmidt-Erfurth et al., 2018). In contrast, DME trajectories in our cohort followed more stable treatment-response curves, allowing linear models to remain effective. This pattern suggests that the advantage of the LLM-based approach scales with trajectory complexity, increasing its utility where forecasting is most challenging.

While OphthaDT demonstrates the effectiveness of generative serialization, our implementation is limited to a specific subset of clinical variables. Expanding the cohort size and integrating optical coherence tomography (OCT) scans into the serialization pipeline is a natural next step that would leverage the architecture's multimodal capabilities.

In conclusion, OphthaDT demonstrates that LLM-based digital twins can effectively capture the longitudinal dependencies of ocular disease. By handling irregular sampling and heterogeneous clinical time-series events without explicit imputation, this approach offers a practical methodology for clinical trajectory forecasting.

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

# A. Methodological Details

## A.1. Prompt Construction

Algorithm 1 details the procedure for constructing the instruction–completion pairs used during supervised fine-tuning. Given the set of all clinical variables $V$, the algorithm selects a forecasting subset $V' \subseteq V$ and produces a text prompt encoding the clinical history and a corresponding target string containing the ground-truth forecasted values.

---

**Algorithm 1** OphthaDT: Prompt Construction for Supervised Fine-Tuning

---

**Require:** Split time $t$, patient history $X_p(t)$, forecasting variables $V' \subseteq V$, targets $Y_p^{\text{forecast}}(t, V')$
**Ensure:** Input prompt $S_{\text{prompt}}$ and target completion $S_{\text{target}}$
 1: Initialize $S_{\text{prompt}}$ with a fixed preamble $\pi_0$.
 2: Encode static baseline data $s_p$.
 3: Serialize longitudinal history $H_p(t)$ chronologically; omit missing values.
 4: If sequence exceeds context window, truncate intermediate visits while preserving baseline.
 5: Encode forecasting tasks as a numbered list of future timepoints.
 6: Construct $S_{\text{target}}$ using ground-truth values in task order.
 7: **return** $(S_{\text{prompt}}, S_{\text{target}})$

---

This prompt is used for supervised fine-tuning, with the training loss masked to compute gradients exclusively on the target tokens $S_{\text{target}}$.

## A.2. Inference and Fine-tuning Details

The OphthaDT training and inference pipeline was implemented using the HuggingFace `transformers` and `trl` libraries. All experiments were conducted on two NVIDIA H100 (80GB) GPUs, with the training process taking approximately 5 hours to reach the optimal checkpoint based on validation loss.

**Training Configuration.** We fine-tuned MedGemma 4B using the AdamW optimizer with a weight decay of 0.1 and a cosine learning rate scheduler (peak learning rate $10^{-5}$). Following standard supervised fine-tuning practice, we applied a loss mask so that gradients were computed exclusively for the target completion tokens.

**Inference and Sampling Logic.** During inference, we differentiated between simulation (generating realistic variety) and forecasting (predicting the most likely trajectory). To obtain stable forecasts, we utilized a temperature $T = 1.0$. For each patient, we generated $M = 10$ independent trajectories with a maximum output length of 120 tokens. This length was empirically selected to cover the necessary BCVA variables. The final prediction was calculated as the element-wise mean of these sampled trajectories to reduce sampling variance.

## A.3. Baseline Hyperparameters

All baseline models were trained using scikit-learn with default hyperparameters. For each model, missing values were addressed via mean imputation and categorical variables were encoded using one-hot encoding. Models were trained independently for each indication (nAMD and DME) and each forecast horizon (weeks 8, 24, 52, and 100).

**Linear Model.** We used ordinary least squares regression (`LinearRegression`) with default parameters.

**Random Forest.** We used `RandomForestRegressor` with 100 estimators, maximum depth 10, all other parameters set to default.

**XGBoost.** We used `XGBRegressor` with 100 estimators, a learning rate of 0.1, subsampling set to 0.8, column sampling set to 0.8, and a maximum tree depth of 6. All other parameters set to default.

## A.4. Synthetic Patient Prompt Example

To illustrate the input–output format of OphthaDT, we provide a synthetic patient example constructed to ensure data privacy compliance.

**Input:**

```
The following is a patient, starting with the demographic data, following
    visit by visit everything that the patient experienced:

Starting with demographic data:
  Patient age is 68 years,
  race is White,
  diagnosis is early nAMD,
  eye laterality is right,
  treatment is treatment arm A (main investigational drug),
  TNFL is yes.
  ...

On the first visit (baseline), the patient experienced the following:
  early nAMD.
  BCVA - visual acuity score is 78.0
  intraocular pressure (mmhg) is 16.0
  center subfield thickness ilm-bm single form (um) is 345.0
  presence of intraretinal fluid is yes.
  heart rate (bpm) is 72.0
  systolic blood pressure (mmhg) is 138.0
  Type 2 Diabetes is diagnosed.
  Cataracts is diagnosed.
  Glaucoma is diagnosed.

4 weeks later, the patient visited and experienced the following:
  drug A 6mg is administered.
  BCVA - visual acuity score is 80.0
  intraocular pressure (mmhg) is 17.0
  center subfield thickness ilm-bm single form (um) is 318.0

  ...

You will now have multiple tasks to complete. Please answer for each task
    in the same order as they are presented.

Task 1 is forecasting:
Your task is to predict the future values of the following variables for
    each cumulative week starting from the last visit:
      bcva - visual acuity score; study the future weeks 4, 8, 12, 16, 20,
      24, 28, 32, 36, 40, 44, 48, 52, 56, 60, 64, 68, 72, 76, 80, 84, 88,
      92, 96, 100
```

**Output:**

```
Task 1 is forecasting:

4 weeks later, the patient visited and experienced the following:
  study eye - bcva - visual acuity score; study is 83.0

8 weeks later, the patient visited and experienced the following:
  study eye - bcva - visual acuity score; study is 82.0

12 weeks later, the patient visited and experienced the following:
  study eye - bcva - visual acuity score; study is 81.0
  ...

100 weeks later, the patient visited and experienced the following:
  study eye - bcva - visual acuity score; study is 76.0
```

## A.5. System Prompt

The following system prompt is prepended to every patient narrative during both fine-tuning and inference, instructing the model on its clinical forecasting role.

As a specialist predictive model in ophthalmology, your task is to forecast the visual acuity trajectory of patients by integrating baseline clinical measurements, treatment history, medical history, adverse events, and any other information provided about the patient.  Use the provided patient data, including BCVA scores, retinal measurements, administered therapies, and comorbidities, to predict all requested tasks.  Deliver precise and clinically relevant predictions to support treatment planning and clinical trial analysis.

