# OpenReview forum: "OphthaDT: Generative Digital Twins for Forecasting Visual Acuity Trajectories in Ophthalmology"
_ICML.cc/2026/Workshop/FMSD — FMSD @ ICML 2026 Poster_

### Official Review · Reviewer_GETU · 2026-05-19
**OphthaDT: Generative Digital Twins for Forecasting Visual Acuity Trajectories in Ophthalmology**

**Rating:** 6
**Confidence:** 4

**Review:**

Summary:

This paper presents OphthaDT, an LLM-based digital twin for ophthalmology that serializes longitudinal multimodal clinical trial data into structured narratives and forecasts best corrected visual acuity (BCVA) over horizons up to 100 weeks. Trained on 3,220 patients from four Phase III trials, OphthaDT achieves the lowest MAE across all timepoints in nAMD and competitive results in DME relative to linear, Random Forest, and XGBoost baselines. The authors argue that OphthaDT’s benefits increase with trajectory complexity and emphasize that the approach handles irregular sampling without explicit imputation.

Pros:

1. Applying text-serialization LLM-based digital twins to ophthalmology is timely and novel, extending recent oncology-focused work to a new clinical domain with different disease dynamics and endpoints.
2. The prompt construction and instruction-following setup are clearly defined, with loss masking and sampling-based inference to improve forecast stability.
3. The approach naturally accommodates heterogeneous events and missingness via narrative serialization without bespoke imputation pipelines.

Cons:

1. Limited baseline breadth: no sequence/time-series or mixed-effects baselines (e.g., LSTM/GRU/Transformer-TFT/Temporal CNN, Gaussian Processes, state-space models, or linear mixed-effects models commonly used for repeated BCVA).

2. Small absolute gains: improvements vs linear models are often tenths of a letter; the clinical meaningfulness of such small MAE deltas is unclear without significance testing.

3. Uncertainty is not quantified (no predictive intervals or calibration), which limits clinical applicability of a generative model.

4. Potential information loss from context truncation and lack of ablation on serialization components.